# Evaluation of LAI Dynamics by Using Plant Canopy Analyzer and Its Relationship to Yield Variation of Soybean in Farmer Field

Shuhei Yamamoto [1], Naoyuki Hashimoto [2] and Koki Homma [1,*]

1   Graduate School of Agricultural Science, Tohoku University, Sendai 9808572, Japan
2   Faculty of Agriculture and Marine Science, Kochi University, Nankoku 7838502, Japan
*   Correspondence: koki.homma.d6@tohoku.ac.jp; Tel.: +81-22-757-4083

**Abstract:** Soybean yield largely varies spatially and yearly in farmer fields. Appropriate growth diagnosis is recommended to stabilize the yield. Leaf area index (LAI) is a representative diagnostic item, but an evaluation method of LAI dynamics with growth has not been established. In this study, we utilized a growth function consisting of an exponential function and a power math function. Parameters were derived from the growth function to be analyzed with yield variability. The LAI was measured weekly by a plant canopy analyzer in farmer fields for 4 years. The dynamics were parameterized by fitting the growth function. The relationship between the parameters of LAI dynamics and soybean yield was analyzed. The growth function was well fitted to measured LAI at $R^2 = 0.82 \sim 0.90$ and RMSE = $0.54 \sim 0.69$ m$^2$ m$^{-2}$. The parameters of the growth function, such as maximum LAI ($LAI_{max}$) and cumulative temperature at maximum LAI ($T_{LAImax}$), quantified the spatial and yearly differences in LAI dynamics, partly explaining those in the yield. The growth function utilized in this study is considered a robust method to quantify LAI dynamics and to diagnose soybean production. The quantification of LAI dynamics may help to develop crop growth monitoring with UAVs (Unmanned Aerial Vehicles) remote sensing as a new diagnostic tool.

**Keywords:** soybean productivity; field investigation; growth diagnosis; growth function

## 1. Introduction

Soybean yield has large variations spatially and yearly in farmer fields. Although several techniques have been developed for improving yield [1–3], yield variability sometimes restricts their application. Yield constraints need to be specified by diagnosing plant growth to apply these techniques. Generally, plant diagnosis includes plant height, leaf area (or leaf area index: LAI), and color. However, the quantitative relationship between diagnostic items and yield still requires clarification.

Because LAI is one of the most important items for diagnosing plant growth, many studies have analyzed its relationship to yield. Sagawa simply reported the significant correlation between maximum LAI and yield [4]. Some studies have suggested that a sufficiently large LAI in the flowering period is important for yield formation [5,6]. The optimum LAI in the flowering period may range from 3.5 to 5 m$^2$ m$^{-2}$ [7–9], corresponding to almost full absorbance of solar radiation [10]. Other studies suggested that sustaining enough LAI at the beginning of the full grain-filling period is important [11,12]. Nakaseko et al. reported that sustaining LAI is effective when LAI is small [13]. The sustained LAI has been expressed as defoliation in some studies. Gazzoni and Moscardi reported that yield decreased due to defoliation at the beginning of the full grain-filling period [14]. Other studies have suggested that the optimum LAI value and the effect of its sustenance and defoliation seem different depending on the region, soil, cultivar, and year [15–18]. These facts indicate that characterizing LAI dynamics, which include the time and value of maximum LAI and defoliation, is important for evaluating soybean growth.

A few studies have quantitatively evaluated LAI dynamics. Watson showed the LAI dynamics of barley, wheat, sugar beets, and potatoes [19]. Hammer et al. estimated the maximum leaf area of grain sorghum based on leaf area dynamics [20]. For soybeans, some studies have measured LAI in farmer fields [21–23], but few studies have evaluated LAI dynamics. The evaluation of LAI dynamics is often restricted because its measurement is laborious. However, Hirooka et al. utilized a plant canopy analyzer and measured LAI dynamics to evaluate the growth environment of rice in farmer fields [24]. Plant canopy analyzers can frequently measure LAI at many places in farmer fields. The application of growth functions is effective for parameterizing LAI dynamics measured by canopy analyzers [24,25].

This study conducted weekly LAI measurements with a plant canopy analyzer for soybeans in farmer fields for 4 years. A growth function was utilized to parameterize LAI dynamics, and the obtained parameters were analyzed with the yearly and spatial variation in soybean yield. Based on the results, an evaluation of LAI dynamics was performed as one of the diagnostic methods to evaluate soybean productivity in farmer fields.

## 2. Materials and Methods

### 2.1. Measurement in Famer's Field

This study was conducted in soybean fields in the coastal area of Sendai city, Miyagi Prefecture, Japan, from 2017 to 2020 (2017: 38°13′34″ N, 140°58′22″ E; 2018 and 2020: 38°13′47″ N, 140°58′58″ E; 2019: 38°14′01″ N, 140°58′41″ E). We selected four adjacent 1 ha fields from dozens of hectares that were managed by an agricultural corporative corporation. Soybeans were planted in converted fields from paddy fields; therefore, planted fields were changed every year: one-year soybean rotation and two-year rice rotation. Planted soybean cultivar was 'Miyagishirome', the most popular cultivar in the prefecture. Basal fertilizer was applied at 200 kg ha$^{-1}$ with '*Daizu senyo ippatsu R 500*' (ZEN-NOH, Tokyo, Japan) in 2017 and 2020 and with compound chemical fertilizer of 14-14-14 in 2018 and 2019. Sowing dates were from 9 to 10 June, 9 to 11 June, 31 May to 1 June, and 8 to 9 June in 2017, 2018, 2019, and 2020, respectively. The sowing density was approximately 5 cm per seed in a 70 cm interval row. The harvest dates were 26 October, 26 October, 29 October, and 27 October in 2017, 2018, 2019, and 2020, respectively. The number of measurement points was 80 (20 for each field) in 2017, 2018, and 2019 and 32 (8 for each field) in 2020. LAI was measured with a plant canopy analyzer, 'LAI-2200' (LI-COR, Inc., Lincoln, NE, USA), once a week after sowing to maturity. Measurements were performed according to Malone et al. [26]. A 90° view cap was attached to the fisheye lens to avoid effects from the measurer. An average of 2 sequences of measurement at the point was used for the analysis. Plants were harvested with pruning scissors for 2 m length in a row, including the measurement point. Yield and its components were determined after threshing and drying the plants at 80 °C for 72 h.

### 2.2. Analysis of LAI Dynamics with a Growth Function

Mathematical functions, such as the exponential function, logistic function, and Gompertz function, are commonly helpful for quantitatively describing the crop growth process [27,28]. Analyzing with functions fitting to observed crop traits data is based on the growth theory of relative growth rate [29] or empirical practices. For example, the exponential function was successfully used to trace early crop growth [28,30,31]. We tested several widely used functions and their combinations to describe whole growth from emergence to maturity. This study utilized a growth function that combined exponential and power math functions (1).

$$L = a \exp{(b\,T)}\,(1 - T/c)^{\,d} \tag{1}$$

where $L$ is the LAI and $T$ is the cumulative temperature (°C). The cumulative temperature is the sum of the daily average temperature minus the basal temperature (8 °C) since the

sowing date. *a*, *b*, *c*, and *d* are coefficients obtained by fitting to the measured LAI. The equation derives the first and second derivatives as Equations (2) and (3), respectively.

$$dL/dT = a\,b\,\exp{(b\,T)}\,(1 - T/c)^{\,d} - a\,d\,(1/c)\,\exp{(b\,T)}\,(1 - T/c)^{\,(d-1)} \tag{2}$$

$$d^2L/dT^2 = a\,b^2\,\exp{(b\,T)}\,(1 - T/c)^{\,d} - a\,b\,d\,(1/c)\,\exp{(b\,T)}\,(1 - T/c)^{\,(d-1)} -$$
$$(a\,b\,d\,(1/c)\,\exp{(b\,T)}\,(1 - T/c)^{\,(d-1)} - a\,d\,(d-1)\,(1/c)^{\,2}\,\exp{(b\,T)}\,(1 - T/c)^{\,((d-1)-1)}) \tag{3}$$

These three equations characterize LAI dynamics (Figure 1). We defined 7 parameters as follows: (1) cumulative temperature at maximum LAI ($T_{LAImax}$) is $T$ at $dL/dT = 0$; (2) maximum LAI ($LAI_{max}$) is $L$ at $T = T_{LAImax}$; (3) cumulative temperature at maximum LAI growth rate ($T_{LGRmax}$) is $T$ at $dL^2/dT^2 = 0$ and $dL/dT > 0$; (4) maximum LAI growth rate ($LGR_{max}$) is $dL/dT$ at $T_{LGRmax}$; (5) LAI at maximum LAI growth rate ($LAI_{LGRmax}$) is $L$ at $T_{LGRmax}$; (6) cumulative temperature when LAI decreases to half of LAImax ($T_{LAIhalf}$) is $T$ at $L = LAI_{max}/2$; and (7) the LAI decreasing rate at $T_{LAIhalf}$ ($LDR_{LAIhalf}$) is $dL/dT$ at $T_{LAIhalf}$.

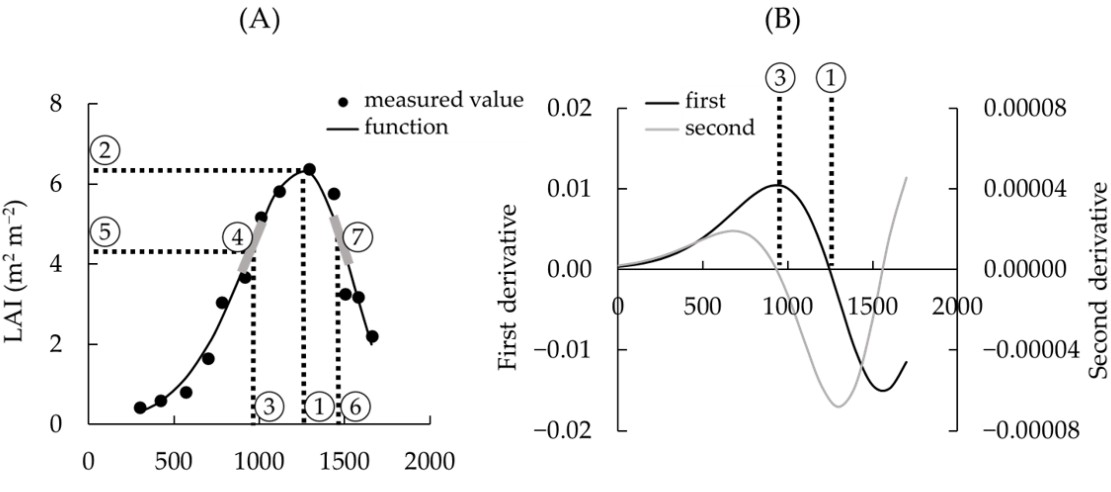

**Figure 1.** Example of (**A**) the application of the growth function on the measured LAI and (**B**) its first and second derivatives. Cited numbers denote the parameters: ① cumulative temperature at maximum LAI ($T_{LAImax}$), ② maximum LAI ($LAI_{max}$), ③ cumulative temperature at maximum LAI growth rate ($T_{LGRmax}$), ④ maximum LAI growth rate ($LGR_{max}$), ⑤ LAI at maximum LAI growth rate ($LAI_{LGRmax}$), ⑥ cumulative temperature when LAI decreased to half of $LAI_{max}$ ($T_{LAIhalf}$), and ⑦ LAI decreasing rate at $T_{LAIhalf}$ ($LDR_{LAIhalf}$).

### 2.3. Statistical Analysis

Fitting Equation (1) was conducted with a solver in Microsoft Excel (Microsoft Corporation, Redmond, WA, USA) to minimize the sum of the squared difference between L and the measured LAI. Analysis of variance was conducted with R version 4.02 (R Core Team, 2020).

## 3. Results

### 3.1. Yield and Yield Components

Table 1 shows the mean and standard deviation of the yield and yield components from 2017 to 2020, varying among measurement points, fields, and years. The ANOVA indicated that the variation of yield among years was significantly larger than that among measurement points ($p < 0.001$). Hundred-grain weight and full seed rate varied among the fields by years (significant interaction). The variation of the two components among fields was obvious in 2017 and 2019. The number of pods closely correlates with yield, but other yield components were weakly correlated.

**Table 1.** Mean and standard deviation of yield and yield components from 2017 to 2020 and the results of the ANOVA. ANOVA tested for variation by the years and the fields against that by the measurement points (within the fields).

| | Yield (g m$^{-2}$) | Number of Pods (m$^{-2}$) | Hundred-Grain Weight (g) | Full Seed Rate |
|---|---|---|---|---|
| 2017 | 162.0 | 405.0 | 36.9 | 0.68 |
| Field | ±51.8 [†] | ±210.0 [†] | ±8.0 *** | ±0.51 *** |
| Point | ±40.9 | ±158.8 | ±2.7 | ±0.17 |
| 2018 | 136.1 | 415.1 | 31.2 | 0.58 |
| Field | ±48.1 [†] | ±50.7 | ±2.3 | ±0.18 * |
| Point | ±34.8 | ±86.9 | ±2.1 | ±0.12 |
| 2019 | 254.7 | 552.6 | 33.9 | 0.66 |
| Field | ±24.5 | ±169.9 | ±9.4 *** | ±0.27 *** |
| Point | ±38.4 | ±144.6 | ±2.8 | ±0.10 |
| 2020 | 256.0 | 714.6 | 35.1 | 0.85 |
| Field | ±27.3 | ±158.1 | ±2.2 * | ±0.058 |
| Point | ±58.7 | ±214.8 | ±1.5 | ±0.052 |
| All Year | 202.2 ± 62.3 | 521.8 ± 145.1 | 34.3 ± 2.4 | 0.69 ± 0.11 |
| Year | *** | *** | *** | *** |
| Field | n.s. | n.s. | *** | *** |
| y × f | n.s. | * | *** | *** |

***: $p < 0.001$, *: $p < 0.05$, [†]: $p < 0.1$, and n.s.: not significant.

### 3.2. Analysis of LAI Dynamics

The growth function was fitted to the measured LAI, resulting in determination coefficients ($R^2$) of 0.90, 0.88, 0.88, and 0.82 and root-mean-squared errors (RMSEs) of 0.69, 0.62, 0.54, and 0.67 m$^2$ m$^{-2}$ on average for the measurement points in 2017, 2018, 2019, and 2020, respectively. The parameters of LAI dynamics based on the function are shown in Table 2. The ANOVA indicated that the variation among fields by years (interactions) was significantly larger than that among measurement points ($p < 0.001$). LAI dynamics in terms of measured LAI on average varied among years (symbols in Figure 2). In 2017 and 2018, the LAI rapidly increased at the early growth stage and rapidly decreased after reaching the maximum. In 2019 and 2020, LAI more slowly increased at the early growth stage and more slowly decreased after reaching the maximum. The measured LAI at the peak was relatively larger in 2017 and 2018, while a larger LAI was sustained at later growth stages in 2019 and 2020. These differences in LAI dynamics were recognized in the parameters in Table 2. For example, LAI$_{max}$ was larger in 2017 and 2018 than in 2019 and 2020, while T$_{LAIhalf}$ was larger in 2019 and 2020 than in 2017 and 2018. The parameter variation per field did not correspond to the yield and yield components in Table 1.

### 3.3. Relationship between Yield and LAI Dynamics

Figure 2 shows the difference in LAI dynamics between the top 10% and bottom 10% yield measurement points. The differences were obvious in 2017 and 2020 but not obvious in 2018 and 2019. These differences may suggest that a larger LAI contributes to higher yields in 2017 and 2020, but other yield constraints limited yields in 2018 and 2019. The relationship between yield and LAI$_{max}$ among measurement points also indicated a yearly difference in LAI contribution to yield; the coefficients of correlation were significant in 2017 and 2020 (Figure 3). Figure 3 also indicated that the difference among the fields was partly obvious in LAI$_{max}$ but not obvious in yield, as shown by the ANOVA in Tables 1 and 2. Although the yearly variation in the parameters of LAI dynamics was quite limited for only 4 years, lower LAI$_{max}$ and larger T$_{LAImax}$ tended to have higher yields (Figure 4). Parameters related to the defoliation, such as LDR$_{LAIhalf}$, did not show a significant correlation with yield (data not shown). The set of parameters of LAI$_{max}$ and T$_{LAImax}$ seemed to relate to the duration of LAI sustenance around LAI$_{max}$ (Figure 2). These results suggested that identifying specific parameters was difficult in relation to the

yield, but the set of parameters helped to reveal the associations between LAI dynamics and yield.

**Table 2.** Mean and standard deviation of the parameters obtained from the growth function from 2017 to 2020 and the result of the ANOVA. ANOVA tested for variation by the years and the fields against that by the measurement points (within the fields). $T_{LAImax}$: cumulative temperature at maximum LAI; $LAI_{max}$: maximum LAI; $T_{LGRmax}$: cumulative temperature at maximum LAI growth rate; $LGR_{max}$: maximum LAI growth rate; $LAI_{LGRmax}$: LAI at maximum LAI growth rate; $T_{LAIhalf}$: cumulative temperature when LAI decreased to half of $LAI_{max}$; and $LDR_{LAIhalf}$: LAI decreasing rate at $T_{LAIhalf}$.

|  | $T_{LAImax}$ (°C) | $LAI_{max}$ (m² m⁻²) | $T_{LGRmax}$ (°C) | LGRmax (m² m⁻²/°C) | $LAI_{LGRmax}$ (m² m⁻²) | $T_{LAIhalf}$ (°C) | $LDR_{LAIhalf}$ (m² m⁻²/°C) |
|---|---|---|---|---|---|---|---|
| 2017 | 1223.4 | 6.5 | 890.5 | 0.011 | 4.1 | 1591.0 | −0.013 |
| Field | ±135.3 *** | ±2.3 *** | ±77.2 *** | ±0.0052 *** | ±1.4 *** | ±209.9 *** | ±0.0057 *** |
| Point | ±42.2 | ±0.92 | ±38.5 | ±0.0021 | ±0.57 | ±60.9 | ±0.0025 |
| 2018 | 1194.8 | 5.8 | 815.6 | 0.0080 | 3.8 | 1606.4 | −0.011 |
| Field | ±120.5 *** | ±1.5 *** | ±127.1 ** | ±0.0025 ** | ±0.90 ** | ±59.9 | ±0.0023 |
| Point | ±60.7 | ±0.76 | ±66.9 | ±0.0015 | ±0.49 | ±50.2 | ±0.0021 |
| 2019 | 1469.3 | 5.21 | 1103.3 | 0.0063 | 3.6 | 1807.9 | −0.016 |
| Field | ±127.8 *** | ±1.9 *** | ±166.7 *** | ±0.0032 *** | ±1.4 *** | ±42.9 ** | ±0.016 *** |
| Point | ±52.5 | ±0.67 | ±63.9 | ±0.0012 | ±0.50 | ±23.6 | ±0.0054 |
| 2020 | 1265.2 | 4.7 | 879.2 | 0.0070 | 3.0 | 1697.7 | −0.0081 |
| Field | ±43.9 | ±1.4 * | ±51.7 | ±0.0032 *** | ±0.87 * | ±136.5 *** | ±0.0037 *** |
| Point | ±44.8 | ±0.87 | ±52.8 | ±0.0017 | ±0.55 | ±68.2 | ±0.0020 |
| All Year | 1030.5 ± 124.1 | 4.4 ± 0.76 | 737.7 ± 125.2 | 0.0064 ± 0.0020 | 2.9 ± 0.48 | 1340.6 ± 99.9 | −0.010 ± 0.0033 |
| Year | *** | *** | *** | *** | *** | *** | *** |
| Field | n.s. | * | † | n.s. | *** | n.s. | *** |
| Y×F | *** | *** | *** | *** | *** | *** | *** |

***: $p < 0.001$, **: $p < 0.01$, *: $p < 0.05$, †: $p < 0.1$, and n.s.: not significant.

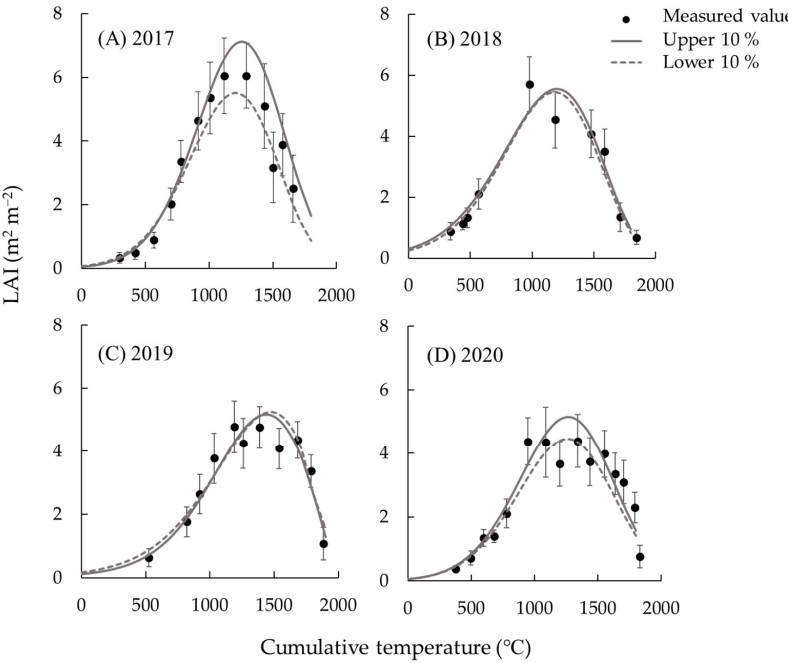

**Figure 2.** LAI value measured with plant canopy analyzer (black circle) and growth functions obtained at the measurement point attaining 10% yield (solid line) and lower 10% yield (break line) from (**A**) 2017 to (**D**) 2020. The error bar is the standard deviation.

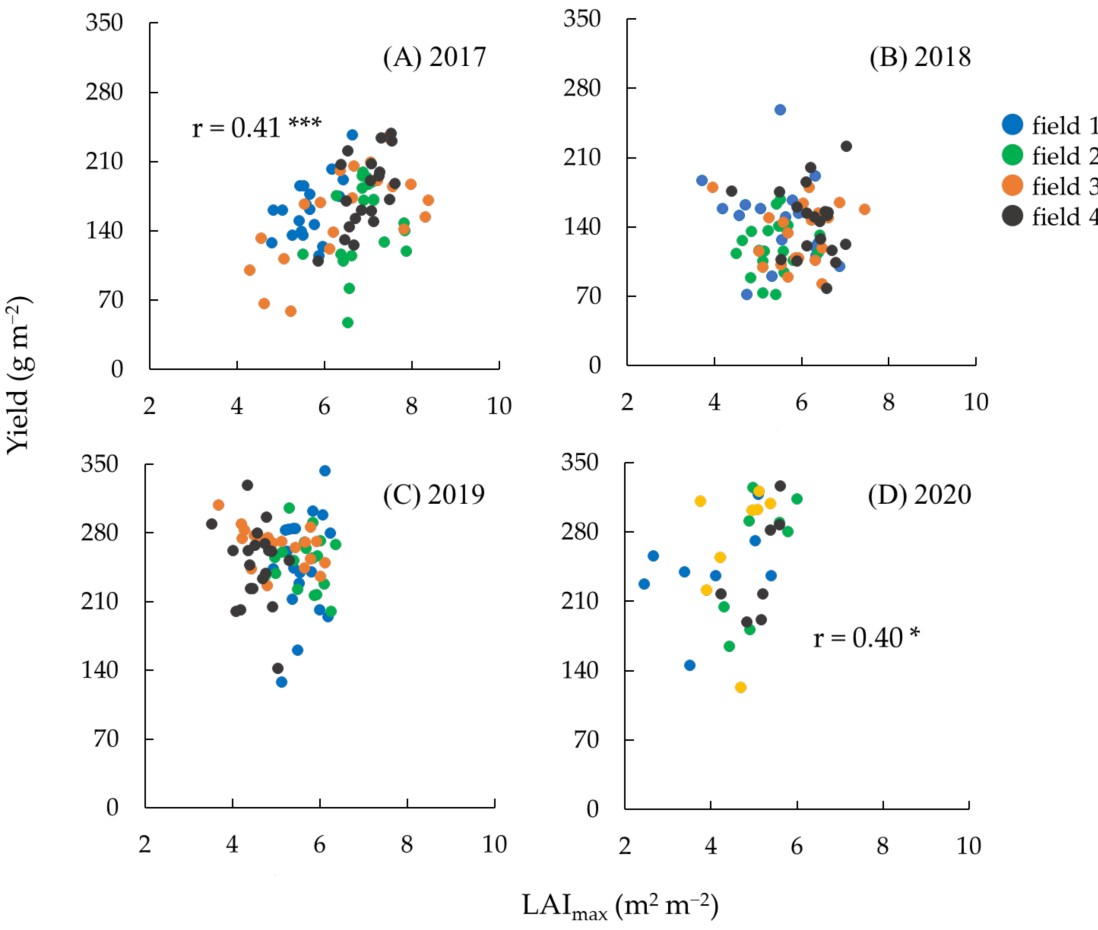

**Figure 3.** Relationship between maximum LAI (LAI$_{max}$) and yield from (**A**) 2017 to (**D**) 2020. r is the coefficient of correlation. ***: $p < 0.001$ and *: $p < 0.05$.

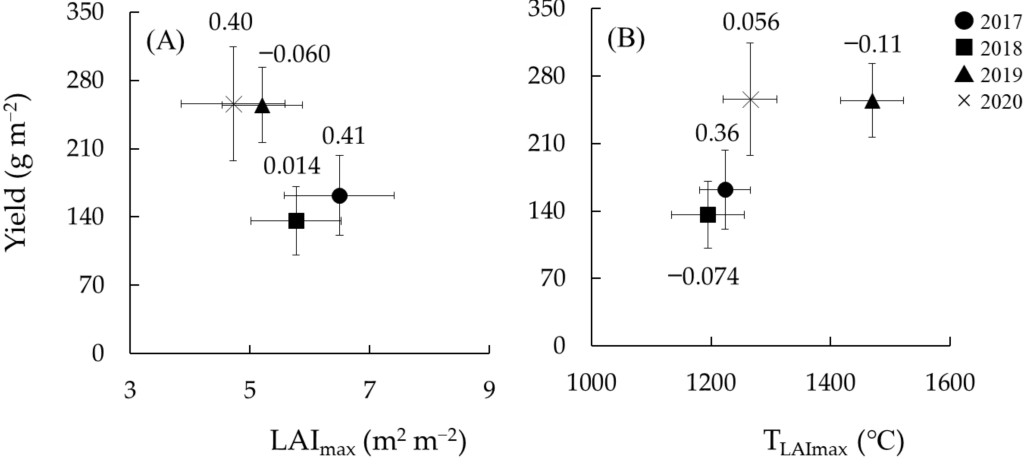

**Figure 4.** Relationship between yield and maximum LAI (LAI$_{max}$, **A**) and cumulative temperature at LAI$_{max}$ (T$_{LAImax}$, **B**). Numbers are correlation coefficients for measurement points in each year. The error bar is the standard deviation.

## 4. Discussion

This study utilized the growth function to analyze the LAI dynamics of soybeans in farmer fields. The function fit the measured LAI relatively well and provided parameters that characterized LAI dynamics. LAI measured by a canopy analyzer substantially contain

measurement error [32,33]. However, the function fitting can statistically reduce the measurement error [32]. We combined exponential and power math functions for the growth function. The exponential function is generally used to express growth in the early growth stage because plants generally show a constant relative growth rate under no constraint conditions [24,34,35]. The power math function is used to express defoliation [36,37], which is generally enhanced at later growth stages. Although several functions have been proposed to express defoliation [38–40], we selected the power math function based on the fitting results (data not shown). The power math function provided two inflection points of the first derivative of the growth function in the case in Figure 1 but only one inflection point in other cases. Accordingly, we defined the parameters, $T_{LAIhalf}$ and $LDR_{LAIhalf}$, at $LAI_{max/2}$ as representatives of defoliation but not the parameters at $d^2L/dT^2 = 0$ and $dL/dT < 0$.

Generally, a larger maximum LAI produced a higher crop yield [4], as partly shown in Figure 2. However, yearly variation in yield had a negative effect on $LAI_{max}$ (Figure 4). The average $LAI_{max} = 6.5$ m$^2$ m$^{-2}$ in 2017 might be excessive and decrease the yield [41], although its inconsistency with the positive correlation between $LAI_{max}$ and yield in 2017 needs further examination. LAI dynamics in 2019 and 2020 show that an adequate LAI (3 to 5 m$^2$ m$^{-2}$) was sustained for a longer period of the growing season. These results suggest that sustaining LAI is more effective for higher yields than maximum LAI. A larger $T_{LAImax}$, which tended to produce a higher yield, indicates that the LAI peaks later in the growth stage. Smaller $LGR_{max}$ and $LDR_{LAIhalf}$ also seem to be important parameters together with larger $T_{LAImax}$ to characterize sustaining LAI. This set of parameters might lead to an optimum LAI, which enhances canopy photosynthesis and increases yield.

As discussed above, no parameters were directly related to the yield variation, although a set of parameters may be associated with productivity. One of the restrictions is probably derived from the shape of the growth function. The function assumes a mountain-shaped curve around $LAI_{max}$, but the shape seems inadequate to represent LAI changes, especially after $LAI_{max}$; the measured LAI seemed to stagnate after its peak in 2019 and 2020 (Figure 2). The restriction can be solved by adding another function, but an increase in the coefficients of another function may decrease robustness. The relatively simple growth function in this study may be considerably better. Another restriction is that the fitting growth function ignores small changes in LAI. Sinclair et al. reported that defoliation is enhanced by the translocation of nitrogen from leaves to grain (self-destruction) [42]. Hirasawa et al. and Neyshabouri et al. reported that LGR decreased with water deficit [43,44]. An accurate evaluation of LAI dynamics may detect such changes in LAI. A strategy to detect LAI change from the expected value needs to be developed.

Fitting crop growth models [45–48] is another strategy to associate LAI dynamics with yield formation. These models include the effects of self-destruction and water deficit on the LAI, suggesting that the causes can be estimated based on LAI dynamics. However, many parameters in the model are difficult to estimate. Reducing or specifying appropriate parameters should be considered [49]. Integration of the growth function in this study into a crop model has the possibility of utilizing measured LAI, although the parameters of LAI dynamics need to be linked with the growth process in the model.

The parameters of LAI dynamics partly explained the observed yield variability but hardly explained the large part of the variability, especially that among the measurement points (within a field; Figures 3 and 4). This implies the necessity to include the effect of other constraint factors. The representative constraints in Miyagi Prefecture are adverse soil in converted fields from paddy fields [50] and diseases [51]. In the study fields, we reported that soil heterogeneity and water excess decreased yield in 2017 [52,53], and harmful soil-borne disease, red crown rot, also did in 2018 and 2020 [51,54]. These constraints might affect LAI dynamics in addition to yield. However, obvious relationships were hardly found between the parameters of LAI dynamics and the factors associated with constraints, such as soil moisture content. We now tried to develop an evaluation method of red crown rot damage based on UAV (Unmanned Aerial Vehicle) remote sensing. The development

of an evaluation method for production constraints is recommended to diagnose soybean production aside from the evaluation of LAI dynamics.

The result of this study also indicated that LAI dynamics sometimes varied even among adjacent fields (Table 2), but the variation did not always cause the difference in yield and yield components (Table 1 and Figure 3). This inconsistency may associate with soil properties, hydrological properties, cropping history, or land use history from the study fields that were reconstructed after the tsunami after the Great East Japan Earthquake in 2011. The reconstruction aggregated several small fields (about 0.3 ha) to a large field (about 1 ha) with a large amount of sediment [55], affecting the relationship between LAI dynamics and yield. To conduct a comprehensive evaluation, including soil properties and management factors, several statistical approaches to treat multivariate would be effective. Mikoshiba et al. showed the effectiveness of covariance structure analysis in evaluating soybean productivity using many physiological and ecological variables [22]. The Bayesian network also enables the analyze crop productivity by connecting many factors [56,57]. A future study requires continuous investigation in farmer fields and evaluation of the relationship between these factors and yield with statistical approaches.

Measuring LAI in farmer fields requires considerable time and labor, limiting data collection. General diagnosis of soybeans includes measuring the main stem length, counting the node of the main stem, and checking the growth stage but does not include LAI measurement by agricultural extension workers in Japan [58]. Because canopy analyzers are not popular due to their cost, the LAI is estimated with visual observation. Recently, UAVs have become popular equipment and have been tested for crop growth diagnosis [59–63]. Because LAI estimation by UAV remote sensing still has a problem with accuracy, especially for full-canopy coverage, further study is required to utilize the growth function in this study. However, parameterization of LAI dynamics with the growth function will contribute to establishing a diagnostic method for soybeans with UAV remote sensing.

## 5. Conclusions

Appropriate growth diagnosis is recommended to stabilize yield variation of soybeans in farmer fields. The LAI is a representative diagnostic item, but its dynamics have not been well evaluated. This study measured the soybean LAI weekly in farmer fields with a plant canopy analyzer and quantified its dynamics by fitting the growth function. The growth function mathematically provides the parameters that characterize the dynamics but is difficult to measure. Although the results in this study suggest that the proposed parameters alone are inadequate to diagnose soybean production, the spatial and yearly variation in soybean yield were partly explained by variations in the parameters of LAI dynamics. This study will contribute to establishing a diagnostic method using UAV remote sensing.

**Author Contributions:** Conceptualization, K.H.; methodology, S.Y., N.H. and K.H.; formal analysis, S.Y.; validation, S.Y.; investigation, S.Y. and N.H.; writing—original draft preparation, S.Y.; writing—review and editing, K.H.; supervision, K.H.; funding acquisition, K.H. All authors have read and agreed to the published version of the manuscript.

**Funding:** This work is partly supported by the JICA-JST SATREPS JPMJSA 1909, JSPS KAKENHI 21H02172, and JST SPRING JPMJSP2114.

**Institutional Review Board Statement:** Not applicable.

**Data Availability Statement:** The data presented in this study are available from the corresponding author on request.

**Acknowledgments:** We thank Sendai Arahama, an agricultural cooperative corporation, for providing the soybean fields for investigation and measurements. We thank all members of the Crop Science Laboratory of the Graduate School of Agricultural Science, Tohoku University.

**Conflicts of Interest:** The authors declare no conflict of interest.

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
