# Peer review of "Evaluation of LAI Dynamics by Using Plant Canopy Analyzer and Its Relationship to Yield Variation of Soybean in Farmer Field"

_agriculture, doi:10.3390/agriculture13030609_

Round 1

Reviewer 1 Report

The entitiled "Evaluation of LAI Dynamics by Using Plant Canopy Analyzer and its Relation to Yield Variation in Farmer Fields" tried to assess the relatonship between LAI and soybean yield. The topic is interesting. 

for specific comments:

1. line 18 the RMSE should have unit.

2. The english correction should be done.

3. improve the quality of all figure such as enlarge the fonts.

Author Response

The entitiled "Evaluation of LAI Dynamics by Using Plant Canopy Analyzer and its Relation to Yield Variation in Farmer Fields" tried to assess the relatonship between LAI and soybean yield. The topic is interesting.

Thank you for your valuable comments. We revised the manuscript based on the comments, surely enhancing the information to the readers.

  1. line 18 the RMSE should have unit.

The same unit was added as LAI (Leaf Area Index): m2 m-2.

  1. The english correction should be done.

We improved whole text with the help of English proofreading service.

  1. improve the quality of all figure such as enlarge the fonts.

Figures were revised.

Reviewer 2 Report

This manuscript try to build statistical relationships between LAI and crop yield based on measurements. While, the manuscript is not prepared well and the innovation is insufficient. In addition, the writing skill needs to be improved. Based on the above fact, I prefer to reject the manuscript. The specific comments are as follows,

1) P18. RMSE = 054 ~ 0.69, miss a decimal point

2) P85. the variables in Eq. (1) need to be re-type, e.g., italic. Similar problems are also seen in the explanations for the equation and in the other equations.

3) Please give the specific theories of Eqs. (1-3), e.g., how these equations be derived?

4) The content in Section 3.3 is insufficient.

Author Response

Comment to the Author

This manuscript try to build statistical relationships between LAI and crop yield based on measurements. While, the manuscript is not prepared well and the innovation is insufficient. In addition, the writing skill needs to be improved. Based on the above fact, I prefer to reject the manuscript. The specific comments are as follows,

Thank you for your profitable comments. The writing was improved by the support of English proofreading service.  We improved whole text and revised the manuscript based on the comments, surely enhancing the information to the readers.

1) P18. RMSE = 054 ~ 0.69, miss a decimal point

Revised.

2) P85. the variables in Eq. (1) need to be re-type, e.g., italic. Similar problems are also seen in the explanations for the equation and in the other equations.

Equation descriptions were revised to italic.

3) Please give the specific theories of Eqs. (1-3), e.g., how these equations be derived?

We added information of theories about analysis with mathematical functions. (L93)

4) The content in Section 3.3 is insufficient.

We added more explanation in Section 3.3. (L184)

Reviewer 3 Report

MAIN COMMENTS

LAI measurements were 80 per field for a total of 320 measurements: which presumably took at least 3-5 hours: how did you avoid drift in LAI data (arising from different sun position). Did you randomize data collection? Conversely (if always the same pattern wasa used) a bias is induce by the data collection order. 

I do not understand: 2 m row were harvested per sampled point (i.e. 160 m per field)? Did you use also an harvester to harvest the total area?

Anyhow it is clear from figures 3 and 4 that 2 m are too small and too much sensitive to local variability and are thus undergoing an excessively high variability. As a consequence also related analyses are debatable. 

Also I guess 2meters have been computed at soil level. I believe it would have been more reasonable to consider a fixed number of plants rather than a fixed distance.   

Authors report results differentiated per year. But it would be interesting to have the results reported also per field. 

The authors analyse the effect of cumulative temperature (as e.g. in figure 2) however the low yield recorded in 2018 is most probably theeffect of other phenomena, as e.g. watering/rain. I believe it would be worth discussing also the relevance of water in the work. 

In general the authros should discuss possible events that in specific years might have affected leaf condition (e.g. storms, insects, stresses,...), thus affecting LAI measurements but not (on in a minor way) affecting yield data. This might useful to better explain deviations and in some cases low correlations.

OTHER COMMENTS:

Please add geographical coordinates of the selected fields

Citation 1 is issing, and citation 44-49 are not cited

Correct "fish eye les" with  "fish eye lens"

Author Response

MAIN COMMENTS

Thank you for your kind evaluation for our manuscript. We improved whole text and revised the manuscript based on the comments, surely enhancing the information to the readers.

LAI measurements were 80 per field for a total of 320 measurements: which presumably took at least 3-5 hours: how did you avoid drift in LAI data (arising from different sun position). Did you randomize data collection? Conversely (if always the same pattern wasa used) a bias is induce by the data collection order.

Firstly, LAI measurement points were 20 per field and total 80 in 2017 ~ 2019, 8 per field and total 32 in 2020 (L 77). We start to measure in the early morning (around 6:30 a.m.) and took about 2-3 hours each measurement day. So, drift in LAI data by sun positon effect seemed to be small.

Secondary, in this paper, we applied the function on the measured LAI data to extract parameters which represents LAI dynamics. The procedure statistically reduces the measurement errors.

I do not understand: 2 m row were harvested per sampled point (i.e. 160 m per field)? Did you use also a harvester to harvest the total area?

Since the description confused the reviewer, we revised the sentences (L82). 2 m row were harvested per sampled point, and 20 sampled points per field, i.e. 40 m per field. The total area except for the sample points were harvested with harvesters of Agricultural Corporative Corporation.

Anyhow it is clear from figures 3 and 4 that 2 m are too small and too much sensitive to local variability and are thus undergoing an excessively high variability. As a consequence also related analyses are debatable.

We added more discussion about yield variability (L253). 2 m was determined to analyze variability within a field. The analysis will contribute field observation with UAV (L280), which mainly focuses spatial variability within a field.

Also I guess 2meters have been computed at soil level. I believe it would have been more reasonable to consider a fixed number of plants rather than a fixed distance.  

We determined the sample area by the distance because crop productivity (e.g. LAI and yield) is usually expressed on the basis of a unit land area. We also counted the number of plants per sampled points, but did not use it in this study.

Authors report results differentiated per year. But it would be interesting to have the results reported also per field.

We added results for the difference among fields (Table 1 and 3) and discussed about variation per field (L266).

The authors analyze the effect of cumulative temperature (as e.g. in figure 2) however the low yield recorded in 2018 is most probably the effect of other phenomena, as e.g. watering/rain. I believe it would be worth discussing also the relevance of water in the work.

We added more discussion about factors causing low yield in 2018 (L253).

In general the authors should discuss possible events that in specific years might have affected leaf condition (e.g. storms, insects, stresses,...), thus affecting LAI measurements but not (on in a minor way) affecting yield data. This might useful to better explain deviations and in some cases low correlations.

We added more discussion about yield constraints in the farmer fields and future study (L253, L266).

OTHER COMMENTS:

Please add geographical coordinates of the selected fields

Added. (L65)

Citation 1 is issing, and citation 44-49 are not cited

Revised.

Correct "fish eye les" with  "fish eye lens"

Revised.

Round 2

Reviewer 2 Report

Im pleased that the detailed revisions and improvements were made by the authors although the different color of revisions is not used in the new version which is difficult to find the revision clearly. In my opinion, there is no major problem in the new version. Yet some minor revisions are also needed:

(1) The discussion is suggested to be strengthened. For instance, the error of Plant Canopy Analyzer (PLC ) is suggested to be added in the discussion, please refer to the following study. (Error Analysis of LAI Measurements with LAI-2000 Due to Discrete View Angular Range Angles for Continuous Canopies, Remote Sensing, 2021)

(2) The same problem with the previous version is the formula typesetting.

After the above revisions, I think the manuscript is prepared for publication.

Author Response

I’m pleased that the detailed revisions and improvements were made by the authors although the different color of revisions is not used in the new version which is difficult to find the revision clearly. In my opinion, there is no major problem in the new version. Yet some minor revisions are also needed:

Thank you for the review. I'm sorry that I might miss to delete the correction history. I revised the manuscript depended on the comments. 

(1) The discussion is suggested to be strengthened. For instance, the error of Plant Canopy Analyzer (PLC ) is suggested to be added in the discussion, please refer to the following study. (Error Analysis of LAI Measurements with LAI-2000 Due to Discrete View Angular Range Angles for Continuous Canopies, Remote Sensing, 2021)

The sentences were add to describe the error of canopy analyzer (L195-197). The suffested study was refered [33]. 2 more references were added to strengthen the discussion [32, 54]. 

(2) The same problem with the previous version is the formula typesetting.

Equation descriptions were revised to italic.

After the above revisions, I think the manuscript is prepared for publication.

Thank you for the suggestions. 

Reviewer 3 Report

the paper has been improved in agreement with my comments, and is thus acceptable for publication

Author Response

the paper has been improved in agreement with my comments, and is thus acceptable for publication

Thank you for the approval.